DOI: 10.1038/s41467-018-06210-4　　**OPEN**

# The shape of watersheds

Timothée Sassolas-Serrayet[1], Rodolphe Cattin[1] & Matthieu Ferry[1]

Since the 1950s river networks have been intensely researched in geosciences and hydrology. This led to the definition of scaling laws that described the organisation of landscapes under fluvial incision and were later explored by statistical physics and fractal mathematics. The emblematic Hack's Law proposes a power-law relationship between watershed area and main stream length. Though extensively documented, a wide range of values is still reported for Hack's parameters. Some authors associate this dispersion to local geologic and climatic conditions. Here based on the analysis of large sets of river basins in various climatic and geological settings, we confirm the geometric similarity of river networks. We demonstrate that basin shape is mostly related to Hack's coefficient and not to the exponent, independently of external forcing such as lithology and pluviometry.

[1] Géosciences Montpellier, Université de Montpellier and CNRS UMR 5243, Montpellier 34095, France. Correspondence and requests for materials should be addressed to T.S.-S. (email: timothee.sassolas-serrayet@umontpellier.fr)

Since the mid-twentieth century, geomorphologists have discovered empirical laws suggesting the existence of invariant properties that describe landscape evolution[1–4]. One of the best-known scaling laws of river network is Hack's Law, which states that the length $L$ of the longest channel of a river basin, measured from outlet to drainage divide, scales with the area $A$ of this basin. This law is written as a power function:

$$L = c A^n \qquad (1)$$

where $c$ and $n$ are often referred to as Hack's coefficient and exponent, respectively.

Many approaches[5–8] including self-affine, energy dissipation or percolation theories focused on the significance of $n$ and its relationship with stream networks fractal dimension, river sinuosity or basin elongation. In contrast, Hack's coefficient has rarely been studied[9] and its significance remains enigmatic.

In his original paper on river catchments in Midwestern United States[1], Hack found that for drainage areas inferior to 100 km², $n = 0.6$ but can be as large as 0.7 in some regions, whereas $c \sim 1.5$ but ranges between 1.1 and 2.7 (for $L$ and $A$ expressed in km and km², respectively). He also interpreted this exponent greater than 0.5 as the result of an increase in basin elongation with the increasing catchment area.

Since this pioneering study, Hack's parameters have been estimated in a variety of contexts. Based on a study of 250 randomly selected worldwide basins with an area ranging from 0.25 to 7,800,000 km², Mueller found that the length–area relationship is best fitted using $n = 0.55$[10]. Furthermore, he proposed that $n$ is not a constant, but rather decreases when basin area increases. He suggested that the value of 0.6 obtained by Hack is only relevant for small (<20,000 km²) catchments and an exponent of 0.46 is more suitable for large basins. From an extensive study based on datasets spanning eleven orders of magnitude in basin area, this last conclusion is questioned by Montgomery and Dietrich[11]. They obtained Hack's coefficient and exponent of 2.02 and 0.49 (for $L$ and $A$ expressed in km and km², respectively), regardless of the catchment size, suggesting a geometric similarity of the drainage network. More recent estimates[12–15] based on regional studies give $n$ and $c$ between 0.45 and 0.7 and 1 and 6, respectively. This wide range of values may be partly due to the spread within dataset or to local variations in climatic or geological properties. It can also be related either to various methods assuming different definitions for $L$ or to the use of a non-homogeneous dataset, which combines stream channels and basins either manually digitised from topographic maps or automatically extracted from digital elevation models (DEM).

While the significance of the exponent $n$ has been the focus of numerous studies, the significance of Hack's coefficient has seldom been addressed and remains an open question. Here, we re-evaluate these two coefficients using a consistent processing of DEM data in various settings, which include mountain ranges and plains as well as semi-arid to humid climates. We specifically focus on the role of the basin shape on Hack's parameters using refinements of established methods. We analyse the morphology of drainage basins in the Bhutan Himalaya, where climate and geology are well established[16–18] and show how the size and the geometry of river drainages influence $c$ and $n$. We propose that Hack's coefficient is only controlled by the basin shape, whereas Hack's exponent depends neither on basin area nor on basin elongation. The comparison with other regions underlines the universal character of these findings[11] and confirms the geometric similarity of rivers network, regardless of lithology, uplift or rainfall.

## Results

**Basins shape gives a physical meaning to data dispersion.** The relationship between $L$ and $A$ we obtain from ca. 22,000 basins from the Bhutan Himalaya is consistent with commonly reported Hack's parameters[2,11] (Fig. 1). Note that for a given basin area the obtained $L$ can vary in a ratio of one to three. This data dispersion—though present in all previous studies—is seldom discussed[2,10–12,14]. The use of Gravelius compactness coefficient[19] (GC) (Fig. 2) to define measures of catchment shape reveals that this dispersion is indeed not related to data precision but rather to basin shape (Fig. 3). In addition, it appears that an increase in basin elongation (increased GC) does not correlate with an increase in catchment area. Whatever the spatial scale considered, catchments exhibit a wide range of shapes with GC ranging between 1.2 and 2.1. In agreement with Montgomery and Dietrich[11], and contrary to Muller's conclusions[10] this suggests that basin elongation is not related to basin size.

**Hack's coefficient is related to basin shape.** Although basin shape varies as a continuum, we divide drainage basins into eight GC class intervals to better assess how this parameter controls $c$ and $n$, from mostly circular (GC = 1.3) to very elongated (GC =

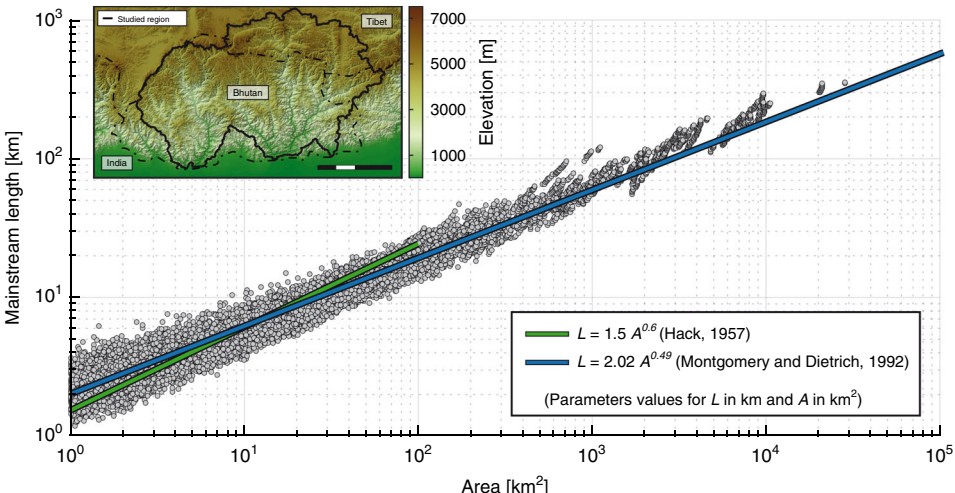

**Fig. 1** 'Length–Area' dataset from ca. 22,000 river basins (grey circles) in Bhutan Himalaya. Green and blue lines show the original Hack's law[2] and the law proposed by Montgomery and Dietrich[11], respectively. Inset shows topographic map of Bhutan region with the limits of the studied region. Scale bar: 100 km

2.0). Except for a few studies[2,13], Hack's parameters are often reported without error bars. Conversely to these traditional approaches focused on the assessment of the best-fitting model, here we favour the use of a likelihood function, which provides information on both robustness and trade-off of these two parameters. Unsurprisingly, the higher the exponent, the lower the coefficient (Fig. 4). The obtained maximum likelihood is in good agreement with parameters reported by Montgomery and Dietrich[11]. Furthermore, the consistency with $c$ and $n$ values proposed by Hack[2] depends on GC. For median values of GC (<1.6) representing about 75% of basins, $c$ and $n$ are in good agreement with Hack's original findings, contrary to what is observed for more elongated basins (GC > 1.6). Considering 95% likelihood to define error bars, our results suggest that $n$ is constant between 0.50 and 0.57 and is independent of the basin shape (Fig. 5a). In contrast, we show that $c$ is clearly influenced by the

basin shape with a relationship between $c$ and GC (Fig. 5b). Virtually, this coefficient corresponds to the mainstream length $L = c$ for a basin of 1 km² (see inset Fig. 3). For the simple case of a rectangular-shaped basin with two sides $a$ and $b$, the relationship between $c$ and GC can be obtained directly from the basin area $A$ and perimeter $P$:

$$A = 1 = ab \Rightarrow b = \frac{1}{a},$$

$$P = 2\sqrt{\pi}\,\mathrm{GC} = 2(a+b)$$
$$= 2\left[a + \frac{1}{a}\right] \Rightarrow a^2 - \sqrt{\pi}\,\mathrm{GC}\,a + 1 = 0, \tag{2}$$

yielding two solutions: $a = \frac{\mathrm{GC}\sqrt{\pi} \pm \sqrt{\mathrm{GC}^2\pi - 4}}{2}$,

Since GC is a not sensitive to the basin orientation with respect to the drainage network, these two solutions represent the upper and lower bounds for the $c$ coefficient (Fig. 5b). Here, the obtained $c$–GC relationship suggests that $c$ is mostly related to rectangle length:

$$c = \frac{1}{2}\mathrm{GC}\sqrt{\pi} + \frac{1}{4}\sqrt{\mathrm{GC}^2\pi - 4}. \tag{3}$$

## Discussion

There is still some debate on how external conditions influence $c$ and $n$. Several studies consider that local conditions associated with lithologic properties, tectonic movements, glaciations and eustatism exert a strong control on the main features of drainage basins[2,12,20] while climatic conditions exert a lower influence[21]. Conversely, the results obtained by Montgomery and Dietrich[11] reveal the ubiquitous character of Hack's exponent. In addition, thermodynamic modelling[22] suggests that external properties have only minor effect on the drainage network because of the small scale of channel initiation compared to the size of geologic domains.

The rainfall distribution in the study area exhibits considerable north–south variations from 0.2 m yr⁻¹ in Tibet to 6 m yr⁻¹ at the Himalayan front[16] (Fig. 6a). We divide our basin dataset into six rainfall classes from dry (<0.5 m yr⁻¹) to very humid (>4 m yr⁻¹) following common practice for the Himalayas[16] and limit our analysis to basins characterised by a single rainfall class to avoid spatial variations that may affect results for large catchments (Fig. 6a). To maintain consistency with our

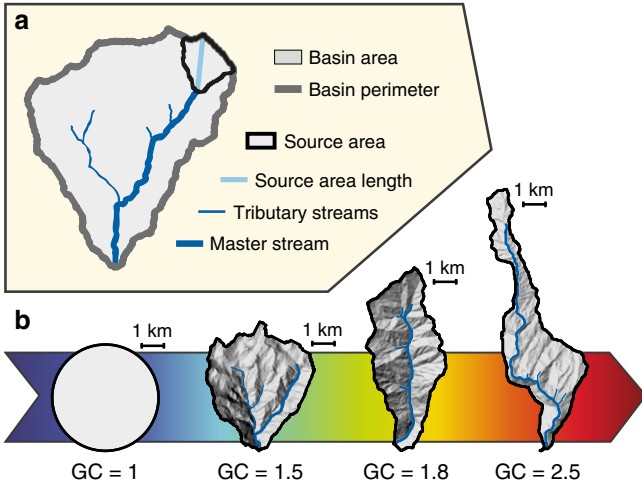

**Fig. 2** Geometric parameters of a drainage network and its watershed basin. **a** Main drainage basin features used in this study. **b** Selected basin shapes from the Bhutan dataset and associated Gravelius coefficients for a constant basin area (ca. 17.5 km²) compared to a perfectly circular basin (GC=1). For each basin, the thick blue line corresponds to the longest stream that defines $L$

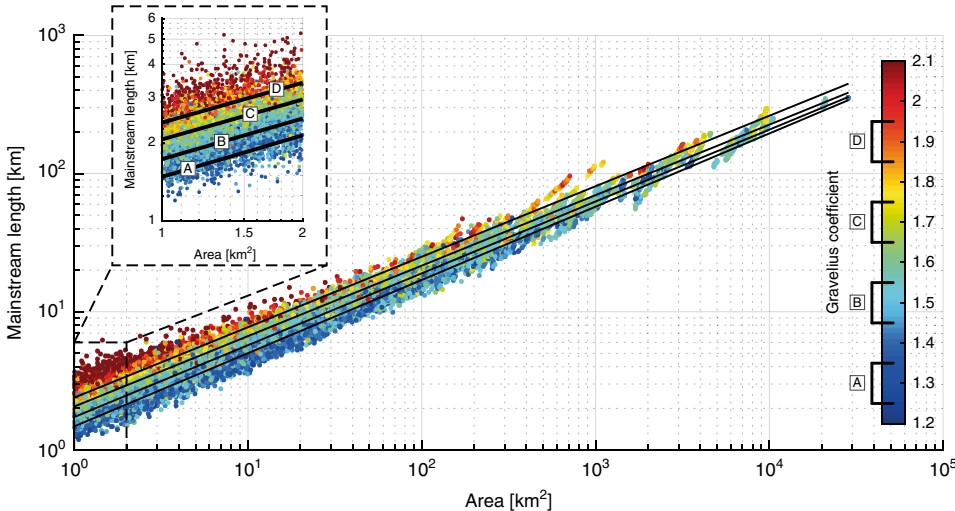

**Fig. 3** 'Length–Area' dataset from ca. 22,000 river basins in Bhutan Himalaya as a function of basin shape. Colour code is for the Gravelius compacity coefficient[19] (GC, see Fig. 2). Inset is a blow-up for basins between 1 and 2 km². Black lines **a**–**d** represent the best fit obtained for subsets based on class intervals of GC [1.25–1.35], [1.45–1.55], [1.65–1.75] and [1.85–195], respectively

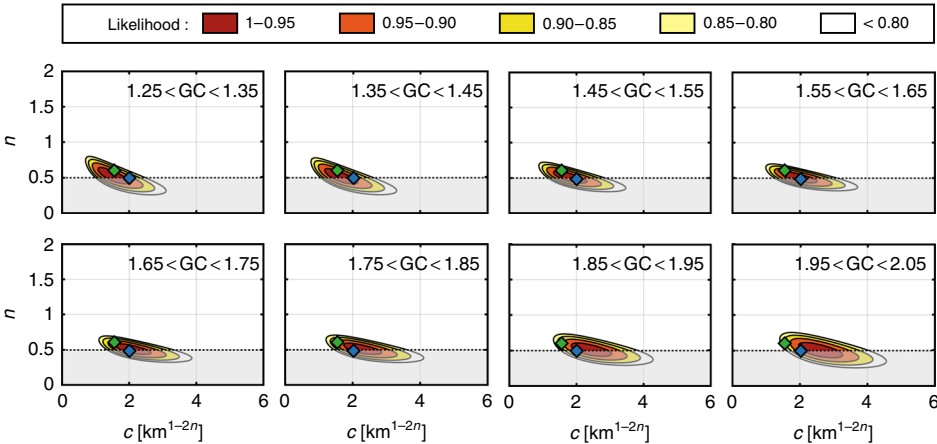

**Fig. 4** Likelihood contour plot of Hack's parameters for eight classes of GC. Colour scale gives the likelihood obtained for each pair of parameters. Green and blue diamond show $c$ coefficient and $n$ exponent as defined by Hack[2] ($c \sim 15$; $n = 0.6$) and Montgomery and Dietrich[11] ($c = 2.02$; $n = 0.49$), respectively. Grey zones correspond to the parameters space that does not satisfy Euclidean geometry and is discarded from our analysis

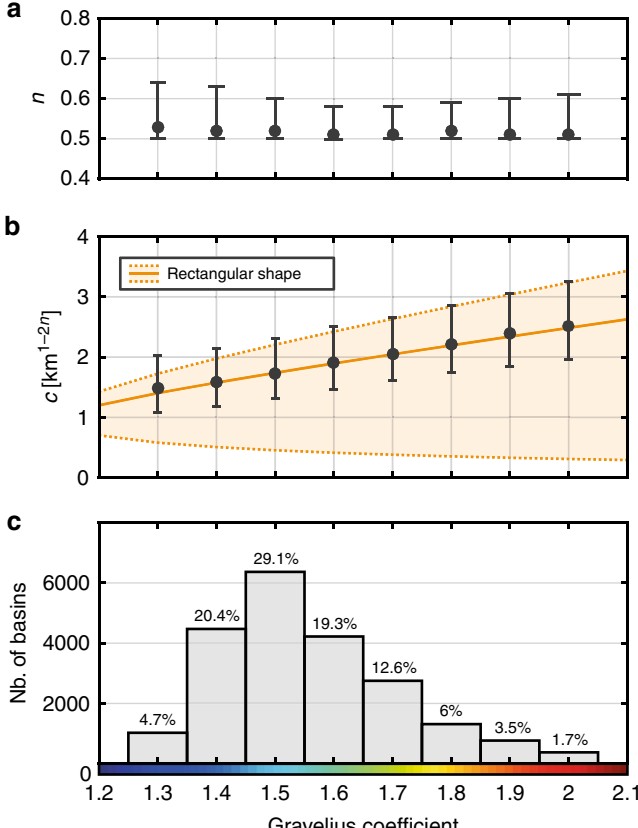

**Fig. 5** Hack's Law parameters vs. Gravelius compacity coefficient.
**a**, **b** Exponent ($n$) and coefficient ($c$) for the eight interval classes of GC showed in Fig. 2. Error bars are defined considering the 95% likelihood contour obtained for each class. Orange solid line corresponds to the relation obtained from Eq. (4). Orange dotted lines represent the lower and upper bounds for the Hack's coefficient assuming a simple rectangular basin. **c** Number of basins for each class of GC and its associated proportion within the dataset

approach, we divide each of the six rainfall classes into eight sub-classes based on GC values (as in Fig. 5). For a given GC class, we observe that both estimated $n$ exponent and $c$ coefficient are independent of the mean annual precipitation (Fig. 7d, f) and

conclude that mean annual rainfall does not play a significant role on Hack's Law.

Similarly, Bhutan can be divided into four distinct major litho-tectonic units encountered along the whole Himalayan arc, which include from north to south: sediments and low-grade metamorphic rocks of the Tethyan Sedimentary Series, high-grade metamorphic rocks of the Greater Himalaya, low-grade metasediments of the Lesser Himalaya and quaternary clastic sediments of the Siwaliks Hills (Fig. 6b). Although these units exhibit significantly different surface erodibility values[23], it appears that their lithologic properties have no major influence on Hack's parameters, regardless of the associated GC class (Fig. 7d, f). The Siwaliks Hills domain crops out along a very narrow strip (<10 km width) along the Himalayan Front in Bhutan and contains only 13 mono-lithology basins (Fig. 6b). Thereby, the strong variability obtained for that litho-tectonic class is not statistically robust and is ignored in our analysis.

To illustrate our approach, we analysed ca. 22,000 sub-basins from four major trans-Himalayan basins and obtain results mostly relevant to these local river networks. To extend the scope of our findings to a wider range of climatic and geologic settings, four additional regions are studied following the same methodology. They include the Pyrenees Ranges to compare with another active belt and South-East Africa, Iowa and coastal Oregon, for which Hack's parameters have been already reported[12,14]. As expected, each area displays its own characteristics with specific $c$ and $n$ parameters (Fig. 8). Note that Hack's parameters for the Pyrenees are very similar to those obtained for Bhutan ($c \sim 1.5-2.5$ and $n \sim 0.5$). Basins from Coastal Oregon have $n \sim 0.5$ identical to exponents obtained in active mountain belts, but with a slightly lower coefficient (1.3–2.2 vs. 1.5–2.5). These results, though significantly different from previously reported parameters ($n = 0.7$ and $c = 1.2$)[12], are derived from a significantly larger dataset (ca. 21,000 vs. 20 basins) and are statistically more robust. This finding leads to a fractal dimension $D = 2n = 1$ for channel length[5] and questions the fractal character of river networks in orogenic zones[24]. This may suggest a similarity of river networks in these specific settings, in spite of very different rainfall and uplift rates.

Compared to Bhutan, basins from Iowa and South-East Africa exhibit higher exponents and lower coefficients. These results are consistent with previous studies[12,14] and suggest a different behaviour for low-relief landscapes. Despite these differences, the comparison with other regions confirms the main findings obtained in

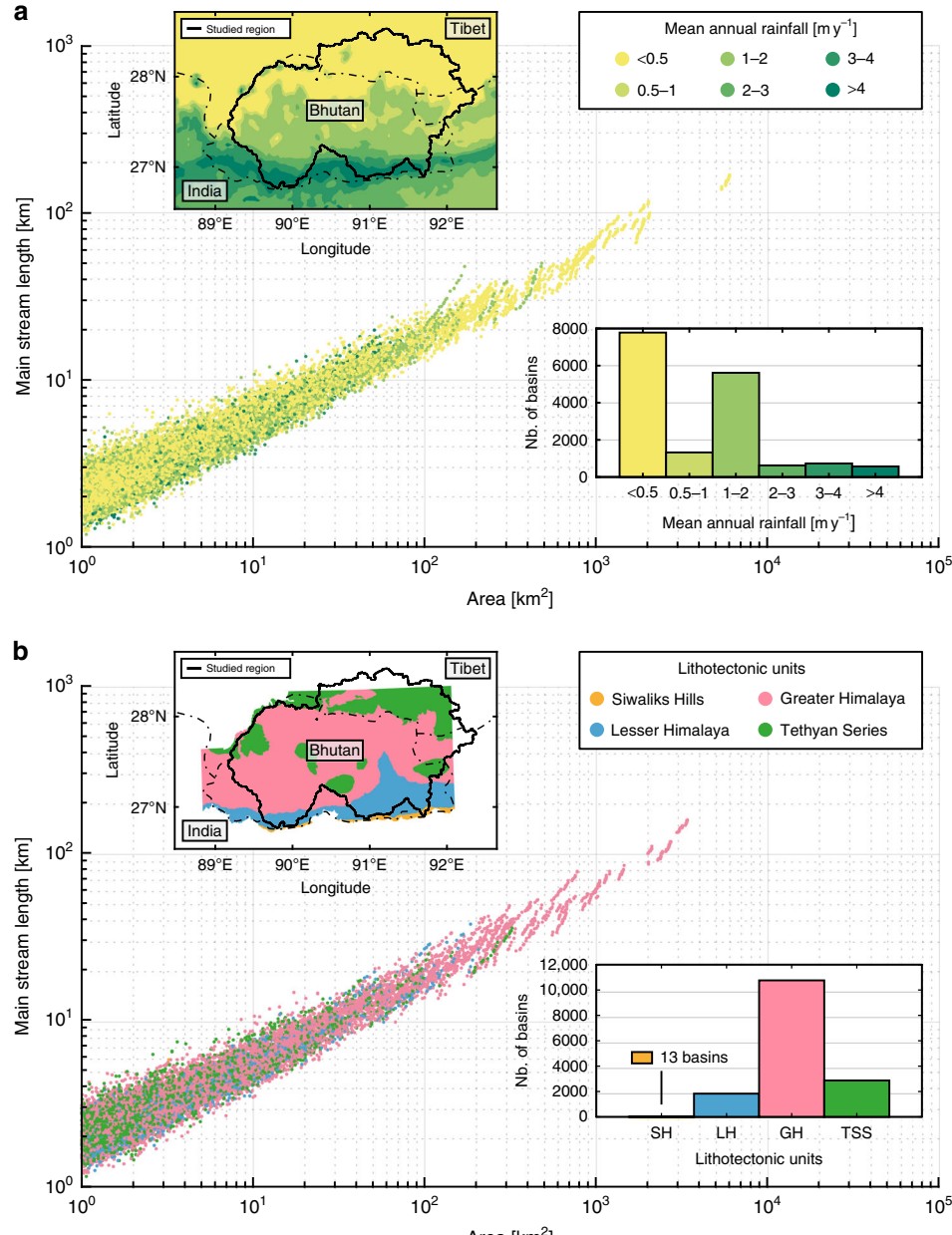

**Fig. 6** Length–area relationship to mean annual rainfall and lithology. **a** Scatter plot of the main stream length with respect to the basin size. Colour scale indicates the annual rate of precipitation for each basin. Inserts are a map of the six main annual rainfall classes[16] and a histogram showing the distribution of basins with respect to these classes. **b** Scatter plot of the main stream length with respect to the basin size. Colour gives the geological domain for each basin. Inserts are a map of the four major litho-tectonic units in Bhutan[17] and a histogram showing the distribution of basins with respect to these units

the Bhutan area. Indeed, basins for all study areas present common significant features, such as a constant Hack's exponent with respect to GC and a basin shape-dependent Hack's coefficient.

The consistency of our results across such very different settings indicates that river basins reach a shape that appears to be unrelated to external influences such as lithology, tectonic uplift or rainfall distribution. Together, these results lead us to confirm the geometric similarity of river networks with basin size-independent Hack's parameters[11]. Our findings reveal that basin shape gives a physical meaning to the data dispersion observed in previously published L–A data sets. This parameter needs to be considered in Hack's Law to better describe the main stream length–area relationship. We hence propose a modification of Hack's law (given by

Eq. (1)):

$$L = c(\varphi) A^n, \qquad (4)$$

where $n$ is a constant, ranging between 0.5 and 0.6, and $c$ is a coefficient, which depends on the basin shape $\varphi$ (for this study, the Gravelius Compacity coefficient).

In summary, when applied to five selected regions of low to high relief, our analysis shows that auto-similarity is confirmed for mountainous areas, and that both $c$ coefficient and $n$ exponent are independent of lithology and rainfall, and finally that Hack's coefficient is related to basin shape.

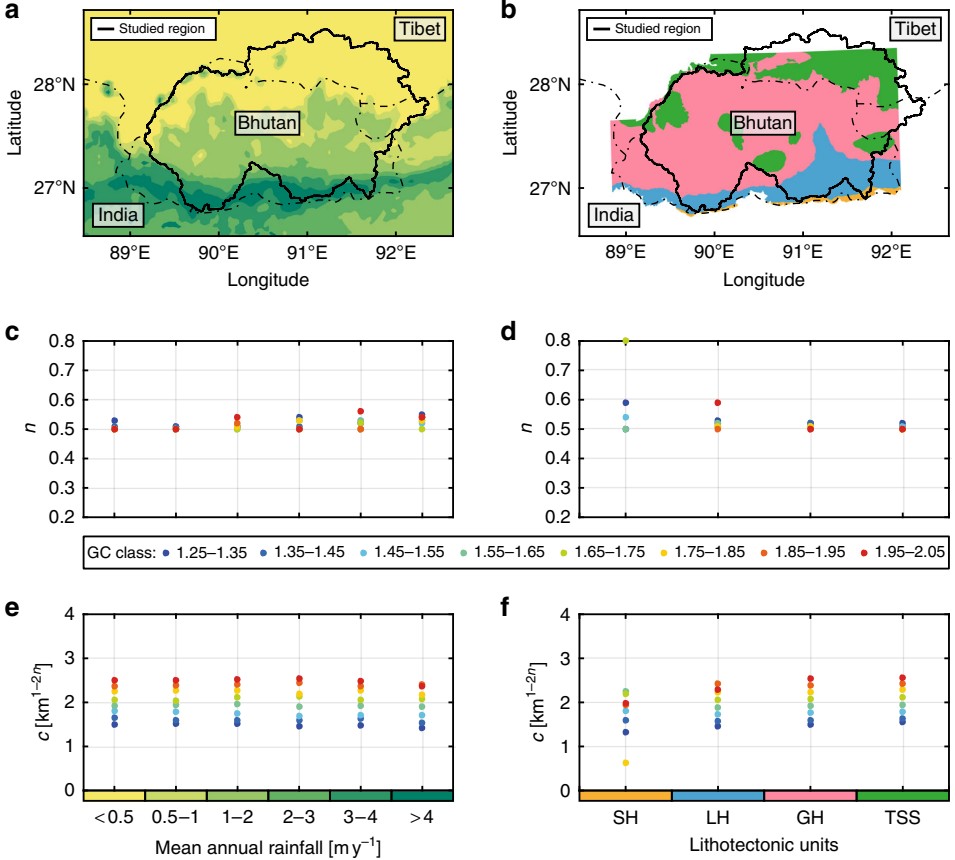

**Fig. 7** Rainfall and lithological effects on Hack's parameters for different classes of basin shapes. **a, b** Maps of the six main rainfall classes[16] and the four major geological units in Bhutan[17]. **c–f** Influence of rainfall (**c, e**) and lithology (**d, f**) on Hack's exponent (**c, d**) and coefficient (**e, f**). Colours of dots indicate the GC class as shown in Fig. 1. Geological units are: Siwalik Hills (SH), Lesser Himalayas (LH), Greater Himalayas (GH) and Tethyan Sedimentary Series (TSS). Hack's exponent shows dispersion of GC values within a given rainfall or lithology class but remains generally constant. Hack's coefficient remains constant for a given GC class regardless of rainfall or lithology. The SH lithology class displays very strong dispersion associated with a scarce dataset (13 basins) and is discarded from our analysis

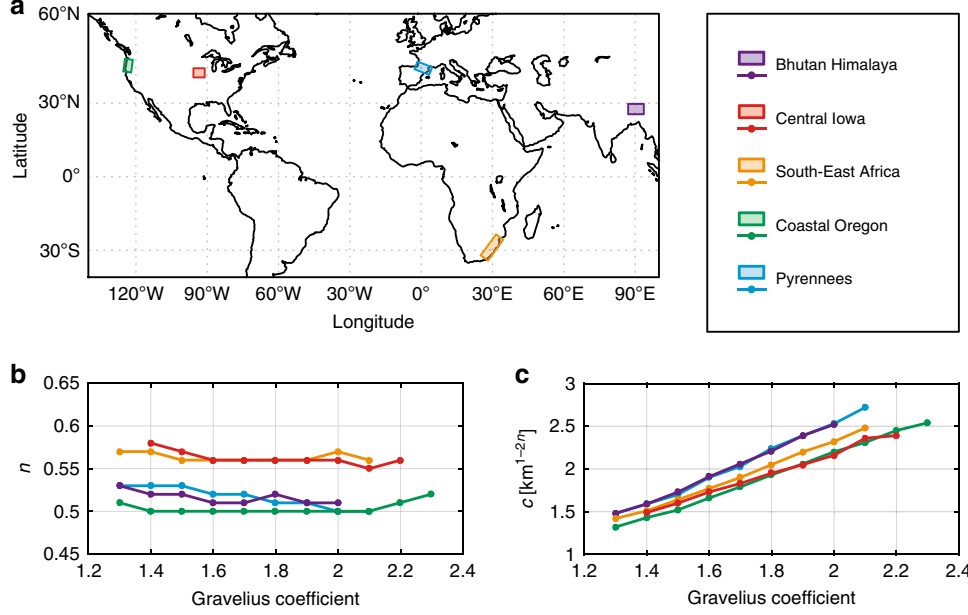

**Fig. 8** Synthesis of results obtained for our global comparison. **a** Map showing the location of the five studied region. **b, c** For all regions, Hack's exponent $n$ (shape-independent) and coefficient $c$ (shape-dependent) exhibit similar trends with respect to the Gravelius Compacity coefficient

## Methods

**Data and study areas**. The topographic dataset used in this study is the 30-m-resolution ALOS—World 3D graciously provided by JAXA. The standard deviation of elevation is estimated to be 5 m[25], which can be considered as negligible with respect to the scale of morphometric properties analysed in this study.

Overall, we applied our approach to five regions in order to provide a wide range of both climatic and geologic settings (Supplementary Fig. 1–5). Areas are chosen for their well-established geology and rates of tectonic activity.

The Bhutan Himalaya is located in the eastern part of the Himalayan arc. This active mountain belt area presents several specific features such as high tectonic uplift (up to 1 cm. yr⁻¹) and high mean annual rainfall (up to 6 m. yr⁻¹) concentrated during the monsoon period. In this study, we consider four distinct tectonic units, which include from north to south, (1) the Tethyan Sedimentary series, (2) the Greater Himalaya, (3) the Lesser Himalaya and (4) the Siwaliks. Here we processed ca. 22,000 sub-basins contained within four major trans-Himalayan basins of the Wang Chhu, Puna Tsang Chhu, Manas Chhu and Bada Chhu.

The Pyrenees are a low tectonic activity mountain range that exhibits moderate rainfall. Its axial zone consists in metamorphic rocks of the Hercynian basement. Results are based on the analysis of ca. 8,500 basins.

The Oregon Coastal Range is an active belt resulting from the subduction of the Juan de Fuca plate. There is no evidence of glaciers during the last glaciation in this region[12]. The lithology is composed of slightly deformed clastic sediments and volcanic rocks. Climate is maritime with an average annual rainfall in the range of 2.5 m yr⁻¹. ca. 21,000 basins are used here.

South-East Africa coast is a passive margin formed 140 Ma ago during the Gondwana break-up. Lithology consists mostly in sedimentary rocks (sandstone, shale and till). The region knows a humid sub-tropical climate with annual rainfall up to 1 m yr⁻¹ around Durban. Its drainage network from the coast to the base of the Drakensberg escarpment has probably not changed over the last 10 million years[14]. Here we used ca. 45,000 basins.

Central Iowa is a tectonically stable continental region, whose plains landscapes were shaped by the last glaciation. Thus lithology is associated with thick layers of glacial and inter-glacial deposits of till, loess and alluvium. Climate is continental with seasonal moderate rainfall. It is suggested that rivers still are in a post-glaciation process of development and basins are not fully evolved yet[12]. About 17,500 basins are studied in this area.

**Main stream length**. We used TopoToolbox[26,27] to derive basins and channel networks from the DEM data. Using flow path computation, stream network is extracted assuming a threshold value of 1 km² source area, which physically represents a transition from steep debris flow-dominated channels to lower-gradient alluvial channels[28]. Sub-basins are defined at each confluence in the drainage network. This results in a great number of sampled basins (several tens of thousands) with a wide size range.

There are many ways of defining the longest stream. The length $L$ is here measured for each basin along the longest valley from the outlet to the divide following the original Hack's method[2], which includes not only the channel length, but also the longest horizontal distance between the location of the stream source and the drainage divide of the source area (Fig. 2a). This prevents problems associated with the definition of channel head position.

**Basin shape**. The shape of a drainage basin is commonly characterised with a dimensionless index, expressed as the ratio between two dimensions of the considered basin. Many indices have been proposed, including the shape factor[1], the elongation ratio[29], the circularity ratio[30] or the aspect ratio[9]. Here we use the Gravelius compactness coefficient GC[19], which is one of the most widely accepted shape indices. It is defined as the ratio between the basin perimeter and the circumference of a circle with a surface equal to the basin area. This coefficient is 1 for an ideally circular watershed and increases with both basin elongation and irregularity of basin boundaries (Fig. 2b). Easy to calculate, GC is well-suited for quantifying the shape of natural drainage catchments, especially for basins with a non-rectangular shape.

The main disadvantage of GC comes from the fractal character of the basin perimeter that causes GC to depend on the DEM resolution. This fractal characteristic also implies an overestimated GC for larger basins due to the presence of small-scale crenulation[31]. To solve both of these issues, we assume that the basin perimeter has to be measured with a relative resolution set by:

$$\text{Rr} = \frac{1}{10}\sqrt{A}, \qquad (5)$$

where $A$ is the area of the basin calculated from the 30-m-resolution DEM[31]. Finally, GC is defined by the equation:

$$\text{GC} = \frac{\text{Pr}}{2\sqrt{\pi \text{Ar}}}, \qquad (6)$$

where Pr and Ar are the perimeter and the area of the basin as calculated with the relative resolution Rr. In that way, GC does not depend anymore on either DEM resolution nor basin scale and becomes a robust shape factor to compare basins

across different scales (Supplementary Fig. 6). We also test the effect of DEM resolution between 30 m and 90 m in our results (Supplementary Fig. 7).

**Error bars**. Here, to assess both uncertainties and possible trade-off between Hack's parameters, we calculate for each GC class interval and for $n$ ranging between 0.5—as expected from Euclidean geometry—and 2, and $c$ ranging between 0 and 6, the likelihood function:

$$\text{Likelihood} = e^{(-\chi^2)},$$
$$\text{with} \quad \chi^2 = \frac{1}{k}\sum\nolimits_{(k=1)}^{k}\left[\log(\text{Lobs}_i) - \log(\text{Lcalc}_i)\right]^2, \qquad (7)$$

where $k$ is the number of basins, and Lobs and Lcalc are the observed and the calculated longest channel, respectively.

## Data availability

The data that support the findings of this sudy are available from the corresponding author on request.

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

## Acknowledgements

T.S.-S.'s Ph.D. is supported by a fellowship from the French Ministry for Higher Education and Research. We thank Wolfgang Schwanghart for providing the programme TopoToolBox to analyse Digital Elevation Models and JAXA for providing the ALOS World 3D data. Authors acknowledge valuable input from two anonymous referees.

## Author contributions

R.C. and M.F. initiated this study. T.S.-S. performed the topographic analyses. All authors contributed to the writing of the paper.

## Additional information

**Competing interests:** The authors declare no competing interests.

