## [Peer Review File · Nature Communications]

Reviewers' comments:

Reviewer #1 (Remarks to the Author):

The paper presents a new analysis of an old problem – the relationship between drainage basin length and area, as characterized by “Hack’s Law”. The paper presents an interesting—and important—new analysis that deserves to be published. But there are issues of presentation, inaccuracy in prior attribution, and interpretation that need to be cleaned up before it should be published. I would encourage the editors to request a thoughtful revision to address these issues as I believe that a revised paper would be a fine contribution.

I will key the following comments to specific lines in the submitted manuscript.

Line 7: The scaling laws (like Hack’s Law) that describe the organization of fluvial landscapes did not arise from “statistical physics” or “fractal mathematics”, as the sentence suggests. These two tools have been used to try and understand or predict the exponents (and less often the coefficients) in such scaling relations, but the laws themselves did not come out of those areas.

Lines 10-11: This statement is not really true. There have been many studies that have constrained Hack’s exponent for different areas (and data sets) and the prior results of studies, and especially the Montgomery and Dietrich (1992) that found the exponent was consistent, and thus self-similar, for a global dataset of basins across a huge range of climate and geological settings. So the expectation from their global result would be that these “explanations” (i.e. geology and climate) would not actually explain much—as the authors of this new paper find. This does not mean that the results presented here are not important—they are—but that the way their importance is presented is a bit misleading.

Line 11-12: I frankly don’t know what to make of a statement that says basin shape controls the relationship between its length and area. Of course it does.

Line 16: The way that the comparison is done the Hack coefficient must be related to basin shape. What is new and convincing here is that the authors’ results confirm the overall self-similar trend in basin geometry reported by Montgomery and Dietrich’s (1992) analysis, and build on that analysis to show that precipitation and lithology do not appear to affect this trend for regional subsets of drainage basins. Specifically, Montgomery and Dietrich (1992, p. 287) report that $n \approx 0.5$ for data across 11 orders of magnitude in basin size and that this indicated that basin shape was scale-independent—the same result reported by Timothée et al., who directly demonstrate that lithology and precipitation do not influence basin shape.

Line 17: the paper does not address basin dynamics and really can’t say anything about “steady-state shape” of basins. What it can say is that basin shape retains self-similarity despite variation in lithology and climate.

Line 21: Here the authors seem to have causality backwards when they suggest that the scaling laws “control the [sic] landscape evolution”. Scaling laws don’t control anything—they emerge from and reflect landscape evolution, not the other way around.

Lines 46-47: Montgomery and Dietrich reported $n=0.49$, and argued that it indicated geometric self-similarity ($n \approx 0.5$) as the generalizable result. The “more recent estimates” the authors refer to are regional studies.

Line 55: After just reviewing the extensive literature on n values it seems rather odd to state that its significance has been “neglected”.

Lines 59-61: The paper does not really “show how the size and geometry of river drainages

influence c and n ." It confirms that size and geometry do not influence n , and geometry, as expressed c simply must be positively correlated with the Gravelius compactness coefficient given the nature of the two things being compared both describing the geometry.

Lines 63-65: This sentence seems to be taking credit for revealing what it is actually confirming from the Montgomery and Dietrich (1992) study.

Lines 69-70: That the "data dispersion" typical of Hack's Law plots is not typically discussed relates back to how this range spans basin shapes from equant to elongated. The colored plot in Figure 1b that shows the high value Gravelius Coefficient basins plot at the upper end of the data dispersion (and low value, equant basins plot at the lower end of the range) is an inevitable, and frankly, trivial geometric result. It does, however, demonstrate that the data dispersion is not due to "data precision".

Line 80: The statement that "basin elongation is mostly related to channel length rather than to basin size" is a doozy. How can basin elongation not be directly related to the length of the longest channel?

Line 93: This is essentially identical to the result of Montgomery and Dietrich (i.e., indistinguishable from $n = 0.5$).

Lines 145-147: If these different areas each have their own specific c and n values, what controls these differences?

Line 152: This finding confirming Montgomery and Dietrich's (1992) conclusion that $n = 0.5$ (and thus that $D = 1$ for channel length) does challenge the fractal character of river length. That is parallel to the analysis for river planform networks by Beauvais and Montgomery (1997). This might be worth a mention here.

Lines 164-166: The assertion that "in contrast to previous works, our findings reveals that the shape of a basin strongly influence the length-area relationship" is misleading for several reasons. First, previous work did not address the role of basin shape on the dispersion of data about the central tendency of a regional (or global) length-area relationship because it is obvious given the geometric relationship between length and area that more elongated basins will have a greater length for their drainage area. The way that the authors have stratified their basins by shape (Gravelius index) makes the result in their Figure 3b inevitable and unenlightening. The statement that "This control gives a physical meaning to data dispersion observed in the previously published L-A data sets" is simply reframing the length-area relationship.

Line 175: The statement that "the c coefficient can no longer be overlooked to capture the physics governing the geometry of river networks" is wildly oversold. First, the c coefficient has not been shown to be related to physics, it is only shown to be related to geometry (which, of course, it must be). The idea that elongated basins plot on the upper bound of the length-area relationship (and equant ones at the bottom) does not tell us anything at all about the physics behind drainage basin evolution. Of course the Gravelius coefficient varies with C (or vice-versa). They are both fundamentally shape related. The comparison shown in Figure 3b is seriously deceptive in that the values shown for "Hack 1957" and for "Montgomery and Dietrich 1992" are for compilations of basins of all shapes, whereas the dots with error bars segregated by Gravelius coefficient are for different shape categories. What these plots show is the basins with more equant shapes (lower Gravelius Coefficient) have c parameters less than for the composite data set of all basin shapes, and that more elongated basins have higher c values than the composite data set. This is simply the result of how the analysis is done, and misrepresents the Hack and Montgomery and Dietrich c values as "predicting" no relationship to the Gravelius Coefficient. Yet what you see on the plots is exactly what you would expect from how the analysis is stratified – the Montgomery and Dietrich (1992) parameters run through the middle of the range of c values that the authors calculate for

the individual subsets of data stratified by basin shape (the Gravelius Coefficient). In other words, the comparison of c values presented in panel b of the figures showing how n and c vary with the Gravelius Coefficient are really spurious comparisons. The pattern of Gravelius coefficients being greater for data points at the high end of the range of values in the plot of Figure 1b simply shows what you would expect, that elongated basin shapes would have high lengths for their drainage area, whereas equant ones would have short lengths for their area (and thus respectively occupy the upper and lower zones of the data scatter for basins of all shapes. The same pattern holds for Figure 4e and 4f, which nicely illustrates that rainfall and lithology do not influence basin shape. But it is a geometric inevitability that the C parameter in the Hack relation will vary with the shape of the basin (as that is what it describes, after all) as characterized by the Gravelius Coefficient.

Line 218: In justifying the threshold value of 1 km² for the stream network delineation the authors mischaracterize what the Montgomery and Foufoula-Georgiou (1993) paper actually showed and said. The authors state that the 1993 paper showed that this threshold value "represents a transition in the dominant sediment transport process from hillslope to channeled valley." What those authors actually proposed is that this size threshold typically represented a transition from "steep debris flow-dominated channels to lower-gradient alluvial channels" (Montgomery and Foufoula-Georgiou, 1993, p. 3925). In other words that transition does not represent the start of channels (as the authors of the submitted manuscript indicate) but a transition from the dominance of hillslope transport processes (defined to include debris flows) to a dominance of fluvial transport processes. Indeed, that this is not the same as the start of a channeled valley is one of the major points of the Montgomery and Foufoula-Georgiou (1993) paper. What the authors of the submitted manuscript are identifying with their 1 km² threshold value is the extent of the fluvial channel network. This bit of terminological confusion should be cleared up in revision. Fortunately, it does not influence the analysis reported in the paper (and is an easy fix).

Figure 6 should appear much earlier in the paper.

I would recommend that the authors reframe points of discussion and presentation to avoid the circular "shape of rivers determines the shape of rivers" arguments that they present. What they have, however, solidly demonstrated is that the self-similar basin geometry reported by Montgomery and Dietrich (1992) based on a global data set compiled from a variety of sources holds up to the standardized analysis the authors present. They also demonstrate well that precipitation and lithologic variability do not affect n and c values in the Hack relationship.

Reviewer #2 (Remarks to the Author):

The article is obviously a thoughtful and useful addition to the discussion of Hack's law. I have only two small "negative" comments. One is, that the Montgomery and Dietrich article cited, asserts that data show that the straight line length of a basin and the area of a basin follow exactly the Euclidean law $A = l^2$, over six or seven orders of magnitude of length, I believe. The interpretation is that basin shape does not influence the Hack's law exponent - though of course it would still influence the prefactor, which is a focus of the present work. A significant advantage of the present work is its treatment of a wide range of data sets and a great deal of data (which was a lack in my publication, where, for example, I had difficulty locating relevant data, and had to use some pretty small data sets).

I do have some theoretical difficulty with Hack's exponents less than 0.5, as the straight line exponent is exactly 0.5, as expected from Euclidean geometry. How is it possible to be straighter than straight? Some comment should be made as to the relevance of such values of the exponent; do they imply a minimum level of uncertainty, for example? The authors suggest an implication that larger basins are more elongated, I guess, but this possibility was allegedly excluded by the Montgomery Dietrich article. Thus, the two comments are related.

Reply to comments

Manuscript NCOMMS-18-09655-T

Reviewer #1:

“The paper presents a new analysis of an old problem – the relationship between drainage basin length and area, as characterized by “Hack’s Law”. The paper presents an interesting—and important—new analysis that deserves to be published. But there are issues of presentation, inaccuracy in prior attribution, and interpretation that need to be cleaned up before it should be published. I would encourage the editors to request a thoughtful revision to address these issues as I believe that a revised paper would be a fine contribution.”

Thank you for this general comment. We addressed all comments and implemented most suggestions of Reviewer #1 to clarify our approach and demonstrate the robustness of our findings.

“Line 7: The scaling laws (like Hack’s Law) that describe the organization of fluvial landscapes did not arise from “statistical physics” or “fractal mathematics”, as the sentence suggests. These two tools have been used to try and understand or predict the exponents

(and less often the coefficients) in such scaling relations, but the laws themselves did not come out of those areas.”

We are agree. Our sentence is not precise enough.

Action: We reorganized the sentence as “*Since the 1950s river networks have been intensely researched in geosciences and hydrology. This led to the definition of scaling laws that described the organization of landscapes under fluvial incision and were later explored by statistical physics and fractal mathematics.*”.

Lines 10-11: This statement is not really true. There have been many studies that have constrained Hack’s exponent for different areas (and data sets) and the prior results of studies, and especially the Montgomery and Dietrich (1992) that found the exponent was consistent, and thus self-similar, for a global dataset of basins across a huge range of climate and geological settings. So the expectation from their global result would be that these “explanations” (i.e. geology and climate) would not actually explain much—as the authors of this new paper find. This does not mean that the results presented here are not important—they are—but that the way their importance is presented is a bit misleading.

Line 11-12: I frankly don’t know what to make of a statement that says basin shape controls the relationship between its length and area. Of course it does.

We agree that values for Hack’s parameters $c=1.78$ and $n=0.49$ obtained by Montgomery and Dietrich (1992) are consistent with a world dataset, which is associated with various climates and lithologies. However, due to data dispersion (ca. up to a factor 3 in basin length L for a given drainage area A) these reported parameters are only based on the assessment of the best-fitting model. Other values of Hack’s parameters would also explain their dataset. This dispersion -though present in all previous studies- may explain the range of Hack’s exponent between 0.45 and 0.7 reported in prior studies (Hack, 1957; Mueller, 1973; Willemin, 2000; Dodds & Rothman, 2000; Walcott and Summerfield; 2009; Shen et al., 2017). As mentioned in our paper, there is still some debate on how external conditions (e.g. lithology, tectonic or climate) influence this data dispersion. Here we propose an alternative explanation, where data dispersion is mostly related to the river basin shape.

Action: To clarify our discourse we modified Lines 10-12 to “*Though extensively documented, a wide range of values is still reported for Hack’s parameters. Some authors associate this dispersion with local geologic and climatic conditions. Here, we propose an alternative approach, in which drainage basin shape is also a key influencing factor in the length-area relationship.*”.

Line 16: The way that the comparison is done the Hack coefficient must be related to basin shape. What is new and convincing here is that the authors’ results confirm the overall self-similar trend in basin geometry reported by Montgomery and Dietrich’s (1992) analysis, and build on that analysis to show that precipitation and lithology do not appear to affect this trend for regional subsets of drainage basins. Specifically, Montgomery and Dietrich (1992, p. 287) report that $n \approx 0.5$ for data across 11 orders of magnitude in basin size and that this indicated that basin shape was scale-independent—the same result reported by Timothée et al., who directly demonstrate that lithology and precipitation do not influence basin shape.

Thank you for these comments. Our results confirm most of Montgomery and Dietrich’s findings and underline the influence of basin shape, with a better control on climatic and lithologic conditions.

Line 17: the paper does not address basin dynamics and really can't say anything about "steady-state shape" of basins. What it can say is that basin shape retains self-similarity despite variation in lithology and climate.

We agree, we do not study temporal evolution of watershed geometry in this paper. We only aim to demonstrate that basin shape does not depend on either local climate or lithology.

Action: we removed "steady state", replaced "geology" by "lithology" and changed "climate" by "pluviometry" in Line 17.

Line 21: Here the authors seem to have causality backwards when they suggest that the scaling laws "control the [sic] landscape evolution". Scaling laws don't control anything—they emerge from and reflect landscape evolution, not the other way around.

We agree, our wording is inaccurate.

Action: we replaced "control" by "describe" in the text (Line 21).

Lines 46-47: Montgomery and Dietrich reported $n=0.49$, and argued that it indicated geometric self-similarity ($n\approx 0.5$) as the generalizable result. The "more recent estimates" the authors refer to are regional studies.

We agree this sentence needs clarification and precision.

Action: We modified the original sentence as "*More recent estimates based on regional studies give n and c between 0.45 and 0.7 and 1 and 6, respectively.*".

Line 55: After just reviewing the extensive literature on n values it seems rather odd to state that its significance has been "neglected".

We agree that our sentence was unclear. Over the last four decades many studies focused on the significance of Hack's exponent. However, except for a few studies (e.g. Hack, 1957; Dodds & Rothman, 2000), Hack's exponents are often reported without errors bar. Hack's coefficient has not been a subject of similar interest and its significance does remain enigmatic.

Action: The text has been rewritten as "*Hence, it appears that in the effort to understand the general meaning of one of the most fundamental empirical laws in geomorphology, the significance of exponent n has been deeply studied, but the robustness of its assessment has been neglected. In parallel, Hack's coefficient has been seldom considered and its significance remains an open question*"

Lines 59-61: The paper does not really "show how the size and geometry of river drainages influence c and n ." It confirms that size and geometry do not influence n , and geometry, as expressed c simply must be positively correlated with the Gravelius compactness coefficient given the nature of the two things being compared both describing the geometry.

We are concerned Reviewer #1 may have misunderstood some aspect of our message. In our approach, we study how the two Hack's parameters are influenced by river geometry including mainstream length, basin area and basin shape. Our study does not only confirm what has been proposed by Montgomery and Dietrich (1992) -that Mueller's results suggesting n varies with basin area may be ruled out- but also that c depends on basin shape. To our knowledge, our study demonstrates for the first time this correlation. Hence, we think that the sentence "... and show how the size and the geometry of river drainages influence c and n " is factual and relevant.

Lines 63-65: This sentence seems to be taking credit for revealing what it is actually confirming from the Montgomery and Dietrich (1992) study.

We agree. Due credit is given line 89, but should appear earlier in this sentence.

Action: We replaced the word “*reveals*” by the word “*underlines*” and added the reference to Montgomery and Dietrich’s (1992) study.

Lines 69-70: That the “data dispersion” typical of Hack’s Law plots is not typically discussed relates back to how this range spans basin shapes from equant to elongated. The colored plot in Figure 1b that shows the high value Gravelius Coefficient basins plot at the upper end of the data dispersion (and low value, equant basins plot at the lower end of the range) is an inevitable, and frankly, trivial geometric result. It does, however, demonstrate that the data dispersion is not due to “data precision”.

This comment sums up one of the main results of our study. Although it may seem obvious from Figure 1b (in the original version of the manuscript) that data dispersion is not due to data precision, no prior study demonstrates this finding. The basin shape is only cited as a factor influencing n and c , as lithology, rainfall or vegetation.

Action: we change “*surprisingly never discussed*” by “*seldom discussed*”.

Line 80: The statement that “basin elongation is mostly related to channel length rather than to basin size” is a doozy. How can basin elongation not be directly related to the length of the longest channel?

We agree this sentence is not informative enough.

Action: We replaced this sentence by “*In agreement with Montgomery and Dietrich, this suggests that basin elongation is not related to basin size*”.

Line 93: This is essentially identical to the result of Montgomery and Dietrich (i.e., indistinguishable from $n=0.5$).

Agreed, as mentioned line 84 and after Reviewer #1’s remark, line 65.

Lines 145-147: If these different areas each have their own specific c and n values, what controls these differences?

Our results suggest lower exponent values for orogenic regions (Coastal Oregon, Pyrenees and Bhutan) regardless of rainfall or uplift rates compared to non-orogenic zones (South-East Africa and Central Iowa). This may be due to the influence of local relief, a lower landscape relief leading to a higher n exponent, and vice versa. A systematic study would be very useful, but it requires additional work that is beyond the scope of the present study.

Line 152: This finding confirming Montgomery and Dietrich’s (1992) conclusion that $n=0.5$ (and thus that $D=1$ for channel length) does challenge the fractal character of river length. That is parallel to the analysis for river planform networks by Beauvais and Montgomery (1997). This might be worth a mention here.

We agree and thank the Reviewer for this apropos reference.

Action: We added a reference to Beauvais and Montgomery’s (1997) study.

Lines 164-166: The assertion that “in contrast to previous works, our findings reveals that the shape of a basin strongly influence the length-area relationship” is misleading for several reasons. First, previous work did not address the role of basin shape on the dispersion of data about the central tendency of a regional (or global) length-area relationship because it

is obvious given the geometric relationship between length and area that more elongated basins will have a greater length for their drainage area. The way that the authors have stratified their basins by shape (Gravelius index) makes the result in their Figure 3b inevitable and unenlightening. The statement that “This control gives a physical meaning to data dispersion observed in the previously published L-A data sets” is simply reframing the length-area relationship.

Although the currently used Hack’s law is only related to L-A datasets, we demonstrate that it can lead to an error up to a factor of 3 on the estimated L from a given basin area A. Here we propose to solve this issue by taking into account an additional parameter related to basin shape in a modified expression of Hack’s Law.

Action: we replaced “*in contrast to previous works, our findings reveals that the shape of a basin strongly influence the length-area relationship*” by “*in contrast to previous works, our findings reveals that the shape of basins needs to be considered in Hack’s Law to better describe the length-area relationship*”.

Line 175: The statement that “the c coefficient can no longer be overlooked to capture the physics governing the geometry of river networks” is wildly oversold. First, the c coefficient has not been shown to be related to physics, it is only shown to be related to geometry (which, of course, it must be). The idea that elongated basins plot on the upper bound of the length-area relationship (and equant ones at the bottom) does not tell us anything at all about the physics behind drainage basin evolution. Of course the Gravelius coefficient varies with C (or vice-versa). They are both fundamentally shape related.

We agree, our approach is based on the geometric features of river drainages and not on physics. Furthermore, after the action associated with previous Reviewer #1’s comment, this sentence is a repetition of lines 163-165.

Action: we removed the concerned sentence.

The comparison shown in Figure 3b is seriously deceptive in that the values shown for “Hack 1957” and for “Montgomery and Dietrich 1992” are for compilations of basins of all shapes, whereas the dots with error bars segregated by Gravelius coefficient are for different shape categories. What these plots show is the basins with more equant shapes (lower Gravelius Coefficient) have c parameters less than for the composite data set of all basin shapes, and that more elongated basins have higher c values than the composite data set. This is simply the result of how the analysis is done, and misrepresents the Hack and Montgomery and Dietrich c values as “predicting” no relationship to the Gravelius Coefficient. Yet what you see on the plots is exactly what you would expect from how the analysis is stratified – the Montgomery and Dietrich (1992) parameters run through the middle of the range of c values that the authors calculate for the individual subsets of data stratified by basin shape (the Gravelius Coefficient). In other words, the comparison of c values presented in panel b of the figures showing how n and c vary with the Gravelius Coefficient are really spurious comparisons. The pattern of Gravelius coefficients being greater for data points at the high end of the range of values in the plot of Figure 1b simply shows what you would expect, that elongated basin shapes would have high lengths for their drainage area, whereas equant ones would have short lengths for their area (and thus respectively occupy the upper and lower zones of the data scatter for basins of all shapes).

The same pattern holds for Figure 4e and 4f, which nicely illustrates that rainfall and lithology do not influence basin shape. But it is a geometric inevitability that the C parameter in the

Hack relation will vary with the shape of the basin (as that is what it describes, after all) as characterized by the Gravelius Coefficient.

We agree, the comparison shown in the concerned figure is not entirely clear. The point was to show that previously reported values from compilations of basins of all shapes are consistent with our results when extracted for an “average” basin shape from our dataset. Concerning the second part of the comment, as we explained in previous answers, this is the first time -to our knowledge- that the correlation between Hack’s coefficient and basin shape is clearly shown.

Action: To avoid any further confusion we removed dashed lines associated with Hack’s (blue) and Montgomery and Dietrich’s (green) parameter values. However, we chose to keep published parameter values in an inset to let the readers make the comparison by themselves.

Line 218: In justifying the threshold value of 1 km^2 for the stream network delineation the authors mischaracterize what the Montgomery and Foufoula-Georgiou (1993) paper actually showed and said. The authors state that the 1993 paper showed that this threshold value “represents a transition in the dominant sediment transport process from hillslope to channeled valley.” What those authors actually proposed is that this size threshold typically represented a transition from “steep debris flow-dominated channels to lower-gradient alluvial channels” (Montgomery and Foufoula-Georgiou, 1993, p. 3925). In other words that transition does not represent the start of channels (as the authors of the submitted manuscript indicate) but a transition from the dominance of hillslope transport processes (defined to include debris flows) to a dominance of fluvial transport processes. Indeed, that this is not the same as the start of a channeled valley is one of the major points of the Montgomery and Foufoula-Georgiou (1993) paper. What the authors of the submitted manuscript are identifying with their 1 km^2 threshold value is the extent of the fluvial channel network. This bit of terminological confusion should be cleared up in revision. Fortunately, it does not influence the analysis reported in the paper (and is an easy fix).

We agree that our justification for a threshold of 1 km^2 was probably not clear enough. This threshold indeed represents a transition from “steep debris flow-dominated channels to lower-gradient alluvial channels” (Montgomery & Foufoula-Georgiou, 1993). Here we choose this value to extract the largest drainage network possible in a domain for which alluvial channel is the dominant mode of sediment transport. Following Hack (1957), the final mainstream length includes not only this channel length, but also the horizontal distance between the location of the stream source defined by this threshold and the drainage divide of the source area. This prevents issues associated with the threshold used to define channel heads.

Action: We replaced the sentence by “*Using flow path computation, stream network is extracted assuming a threshold value of 1 km^2 source area, which physically represents a transition from steep debris flow-dominated channels to lower-gradient alluvial channels*”.

Figure 6 should appear much earlier in the paper.

Action: Figure 6, originally called in the “Method” section, becomes Figure 2 in the revised version, and appears at the beginning of the “Results” section. This makes figure 1b more understandable. Furthermore, we divide the original Figure 1 into two separate panels (originally a and b) into revised Figure 1 and revised Figure 3. This change improves the structure and readability of the manuscript. This does not impact format guidelines set by Nature Communications.

I would recommend that the authors reframe points of discussion and presentation to avoid the circular “shape of rivers determines the shape of rivers” arguments that they present. What they have, however, solidly demonstrated is that the self-similar basin geometry reported by Montgomery and Dietrich (1992) based on a global data set compiled from a variety of sources holds up to the standardized analysis the authors present. They also demonstrate well that precipitation and lithologic variability do not affect n and c values in the Hack relationship.

We agree this would benefit the quality of our conclusions. Our analysis (1) confirms the self-similarity of drainage networks, (2) demonstrates that both Hack’s parameters are unrelated to external influences such as lithology, tectonic uplift or rainfall distribution, (3) underlines the effect of basin shape on the L-A relationship and (4) shows that only coefficient c is a basin-shape-dependent parameter.

Action: We slightly rewrote the conclusion to insist on the “self-similarity/precipitation/lithology” parts of the discussion to end on a stronger note.

Reviewer #2:

The article is obviously a thoughtful and useful addition to the discussion of Hack's law. I have only two small "negative" comments. One is, that the Montgomery and Dietrich article cited, asserts that data show that the straight line length of a basin and the area of a basin follow exactly the Euclidean law $A = L^2$, over six or seven orders of magnitude of length, I believe. The interpretation is that basin shape does not influence the Hack's law exponent - though of course it would still influence the prefactor, which is a focus of the present work. A significant advantage of the present work is its treatment of a wide range of data sets and a great deal of data (which was a lack in my publication, where, for example, I had difficulty locating relevant data, and had to use some pretty small data sets).

I do have some theoretical difficulty with Hack's exponents less than 0.5, as the straight line exponent is exactly 0.5, as expected from Euclidean geometry. How is it possible to be straighter than straight? Some comment should be made as to the relevance of such values of the exponent; do they imply a minimum level of uncertainty, for example?

Thank you for this positive review. We agree the exponent n must be equal to or greater than 0.5, as expected by Euclidean geometry.

Action: In the revised version, we reassess for all study areas the best fitting power law assuming a minimum exponent n of 0.5. Obviously, this action only impacts results for which $n < 0.5$ and limits the lower part of uncertainties to 0.5. Hence, we modified related mentions in the main text, figures 2, 3, 4 and 5 (see below) and captions accordingly. These changes improve the accuracy of our analysis without affecting either results or conclusions.

The authors suggest an implication that larger basins are more elongated, I guess, but this possibility was allegedly excluded by the Montgomery Dietrich article. Thus, the two comments are related.

It seems there is some confusion likely originating from the way “compactness” is defined after the Gravelius Coefficient, i.e. that GC is minimal for equant (i.e. highly compact) basins and maximal for elongated basins. Actually, we clearly state that “an increase in basin elongation (increased GC) does not correlate with an increase in catchment area” (Lines 77-78) and

unequivocally confirm the interpretation originally proposed by Montgomery and Dietrich (1992).

Action: Following Reviewer #1's suggestion, we bring up Figure 6 as Figure 2 to explain how GC is defined and how it relates to shape (or compacity); this should lift any confusion about GC.

REVIEWERS' COMMENTS:

Reviewer #1 (Remarks to the Author):

The revised version of this paper is much improved. But there are a few issues remaining with the paper – most of which can be easily addressed by the authors.

The first general comment I should make is that the title is inappropriate. The paper is not about the shape of rivers. It is about the shape of river basins. The title should be revised accordingly.

Lines 12-14: This sentence "Here, we propose an alternative approach, in which drainage basin shape is also a key influencing factor in the length-area relationship" really says nothing more than shape influences shape. This sentence should simply be cut from the paper. The abstract reads fine without it. And it is perfectly obvious that the shape of drainage basins influences their length-area relationship. For an individual basin this is, of course, trivial, a simple matter of geometry. For a composite relation among numerous drainage basins (what Hack's Law is) it is just as obvious that elongated basins will plot at the high end of the length range for a given drainage area (i.e., at the top of the dispersion in the data set) and that equant ones will plot at the bottom (i.e., at low lengths for a given drainage area). What the paper shows is that basin shape explains the dispersion of data in length-area plots and that the overall relationship is self-similar ($n \approx 0.5$).

Line 20: "the landscape evolution" is not English syntax. Should reword so it reads "...properties that describe landscape evolution."

Line 42: suggest changing "a comparably extensive" to read "an extensive" because the two studies referred to did not have a comparable amount of data for the simple reason that the Montgomery and Dietrich study included all of the data from the Mueller study (and a lot more).

Lines 54-56: the statement that "the significance of exponent n has been deeply studied, but the robustness of its assessment has been neglected" is perplexing. What does the "robustness of its assessment" mean? Moreover, with much attention having been paid to n the statement does not ring true. I would suggest just simplifying the two sentences on these lines to read "While the significance of the exponent n has been addressed in numerous studies, the significance of Hack's coefficient has seldom been addressed and remains an open question."

Lines 70-73: This sentence beginning with the word "Hence, " implied a direct connection with the prior sentence. But the logic is lost on me. The authors seem to be confusing the range of dispersion of the values in c reported around the world with that of an overall or global relationship. And I don't recall Mueller concluding that c varied with basin size (only n). I suggest that the authors simply cut the last two sentences from this paragraph (as they don't really add anything and confuse some issues) and that they then simply continue on with the sentence that begins line 74 to continue the paragraph.

Lines 163-167: these sentences can be clarified and greatly simplified to read: "Our findings reveal that basin shape gives a physical meaning to the data dispersion observed in previously published L-A data sets."

Lines 176-178: The last sentence of the manuscript is pointless and should be omitted. A better ending sentence would be to state that "Our findings unequivocally demonstrate that Hack's coefficient is related to basin shape."

As I indicated in my review of the original manuscript submission the comparison of c values with those reported by Hack and Montgomery and Dietrich in Figure 5 is seriously misleading as the prior values are for composite data sets, whereas those reported in the new paper are for data

binned by Gravelius coefficient (shape) values. This is seriously misleading as it is presented in the figure and the boxes referring to those prior studies should be removed as their presence invites the reader to draw an erroneous apples vs. oranges comparison. The data presented by the authors can stand on its own in this figure.

As a final note, the capitalization of paper titles in the reference list is not treated consistently.

Reply to comments

Manuscript NCOMMS-18-09655A

The shape of watersheds

Timothée Sassolas-Serrayet^{1*}, Rodolphe Cattin¹, Matthieu Ferry¹

¹ Géosciences Montpellier, Université de Montpellier and CNRS UMR 5243, 34090 Montpellier, France.* e-mail: timothee.sassolas-serrayet@umontpellier.fr

Reviewer#1

The revised version of this paper is much improved. But there are a few issues remaining with the paper – most of which can be easily addressed by the authors.

Thank you for this comment. We implemented all the suggestions of Reviewer#1 in order to provide the most suitable revised version of our paper.

The first general comment I should make is that the title is inappropriate. The paper is not about the shape of rivers. It is about the shape of river basins. The title should be revised accordingly.

We agree that the title must be revised.

Action: We change the title to “The shape of watersheds”

Lines 12-14: This sentence “Here, we propose an alternative approach, in which drainage basin shape is also a key influencing factor in the length-area relationship” really says nothing more than shape influences shape. This sentence should simply be cut from the paper. The abstract reads fine without it. And it is perfectly obvious that the shape of drainage basins influences their length-area relationship. For an individual basin this is, of course, trivial, a simple matter of geometry. For a composite relation among numerous drainage basins (what Hack’s Law is) it is just as obvious that elongated basins will plot at the high end of the length range for a given drainage area (i.e., at the top of the dispersion in the data set) and that equant ones will plot at the bottom (i.e., at low lengths for a given drainage area). What the paper shows is that basin shape explains the dispersion of data in length-area plots and that the overall relationship is self-similar ($n \approx 0.5$).

We agree, this sentence may be misleading.

Action: We remove the concerned sentence.

Line 20: “the landscape evolution” is not English syntax. Should reword so it reads “...properties that describe landscape evolution.”

Action: We reword by “properties that describe landscape evolution”.

Line 42: suggest changing “a comparably extensive” to read “an extensive” because the two studies referred to did not have a comparable amount of data for the simple reason that the Montgomery and Dietrich study included all of the data from the Mueller study (and a lot more).

Action: We implemented the correction suggested by Reviewer#1 and reworded this sentence by “an extensive”.

Lines 54-56: the statement that “the significance of exponent n has been deeply studied, but the robustness of its assessment has been neglected” is perplexing. What does the “robustness of its assessment” mean? Moreover, with much attention having been paid to n the statement does not ring true. I would suggest just simplifying the two sentences on these lines to read “While the significance of the exponent n has been addressed in numerous studies, the significance of Hack’s coefficient has seldom been addressed and remains an open question.”

We agree, these two sentences may be simplified into one.

Action: We replaced the two sentences by “While the significance of the exponent n has been the focus of numerous studies, the significance of Hack’s coefficient has seldom been addressed and remains an open question.”

Lines 70-73: This sentence beginning with the word “Hence, “ implied a direct connection with the prior sentence. But the logic is lost on me. The authors seem to be confusing the range of dispersion of the values in c reported around the world with that of an overall or global relationship. And I don’t recall Mueller concluding that c varied with basin size (only n). I suggest that the authors simply cut the last two sentences from this paragraph (as they don’t really add anything and confuse some issues) and that they then simply continue on with the sentence that begins line 74 to continue the paragraph.

We agree.

Action: We removed the two last sentences of the paragraph.

Lines 163-167: these sentences can be clarified and greatly simplified to read: “Our findings reveal that basin shape gives a physical meaning to the data dispersion observed in previously published L-A data sets.”

Action: We changed the concerned sentence by “Our findings reveal that basin shape gives a physical meaning to the data dispersion observed in previously published L-A data sets.” and added the sentence “This parameter needs to be considered in Hack’s Law to better describe the main stream length-area relationship.” to stay consistent with the end of the paragraph.

Lines 176-178: The last sentence of the manuscript is pointless and should be omitted. A better ending sentence would be to state that “Our findings unequivocally demonstrate that Hack’s coefficient is related to basin shape.”

We agree, we have to end on a stronger note concerning the relationship between Hack’s coefficient and basin shape.

Action: We removed this last sentence and replaced it by “(iii) that Hack’s coefficient is related to basin shape.”

As I indicated in my review of the original manuscript submission the comparison of c values with those reported by Hack and Montgomery and Dietrich in Figure 5 is seriously misleading as the prior values are for composite data sets, whereas those reported in the new paper are for data binned by Gravelius coefficient (shape) values. This is seriously misleading as it is presented in the figure and the boxes referring to those prior studies should be removed as their presence invites the reader to draw an erroneous apples vs. oranges comparison. The data presented by the authors can stand on its own in this figure.

Action: We removed both of these boxes which refer to those prior studies.

As a final note, the capitalization of paper titles in the reference list is not treated consistently.

Action: We fixed it.